# Ischemic Preconditioning Improves Handgrip Strength and Functional Capacity in Active Elderly Women

**DOI:** 10.3390/ijerph19116628

**Published:** 2022-05-29

**Authors:** Luiz Guilherme da Silva Telles, François Billaut, Gélio Cunha, Aline de Souza Ribeiro, Estêvão Rios Monteiro, Ana Cristina Barreto, Luís Leitão, Patrícia Panza, Jeferson Macedo Vianna, Jefferson da Silva Novaes

**Affiliations:** 1Physical Education and Sports Department, Federal University of Rio de Janeiro, Rio de Janeiro 21941-901, Brazil; guilhermetellesfoa@hotmail.com (L.G.d.S.T.); profestevaomonteiro@gmail.com (E.R.M.); jeffsnovaes@gmail.com (J.d.S.N.); 2Estácio de Sá University (UNESA), Rio de Janeiro 20261-063, Brazil; geliocunha5@gmail.com; 3Department of Kinesiology, Laval University, Quebec, QC G1V 0A6, Canada; francois.billaut@kin.ulaval.ca; 4Physical Education and Sports Department, Federal University of Juiz de Fora, São Pedro 36036-900, Brazil; alinevalencaedfisica@gmail.com (A.d.S.R.); paty_panza@yahoo.com.br (P.P.); jeferson.vianna@ufjf.edu.br (J.M.V.); 5Department of Physical Therapy, University Center of Augusto Motta of UNISUAM, Rio de Janeiro 21041-020, Brazil; 6Celso Lisboa University Center, Rio de Janeiro 20950-092, Brazil; educacaofisica@celsolisboa.edu.br; 7Sciences and Technology Department, Superior School of Education of Polytechnic Institute of Setubal, 2910-761 Setúbal, Portugal; 8Life Quality Research Centre, 2040-413 Rio Maior, Portugal

**Keywords:** ischemic preconditioning, muscle strength, functional capacity, elderly, handgrip strength

## Abstract

Background: Aging decreases some capacities in older adults, sarcopenia being one of the common processes that occur and that interfered with strength capacity. The present study aimed to verify the acute effect of IPC on isometric handgrip strength and functional capacity in active elderly women. Methods: In a single-blind, placebo-controlled design, 16 active elderly women (68.1 ± 7.6 years) were randomly performed on three separate occasions a series of tests: (1) alone (control, CON); (2) after IPC (3 cycles of 5-min compression/5-min reperfusion at 15 mmHg above systolic blood pressure, IPC); and (3) after placebo compressions (SHAM). Testing included a handgrip isometric strength test (HIST) and three functional tests (FT): 30 s sit and stand up from a chair (30STS), get up and go time (TUG), and 6 min walk distance test (6MWT). Results: HIST significantly increased in IPC (29.3 ± 6.9 kgf) compared to CON (27.3 ± 7.1 kgf; 7.1% difference; *p* = 0.01), but not in SHAM (27.7 ± 7.9; 5.5%; *p* = 0.16). The 30STS increased in IPC (20.1 ± 4.1 repetitions) compared to SHAM (18.5 ± 3.5 repetitions; 8.7%; *p* = 0.01) and CON (18.5 ± 3.9 repetitions; 8.6%; *p* = 0.01). TUG was significantly lower in IPC (5.70 ± 1.35 s) compared to SHAM (6.14 ± 1.37 s; −7.2%; *p* = 0.01), but not CON (5.91 ± 1.45 s; −3.7%; *p* = 0.24). The 6MWT significantly increased in IPC (611.5 ± 93.8 m) compared to CON (546.1 ± 80.5 m; 12%; *p* = 0.02), but not in SHAM (598.7 ± 67.6 m; 2.1%; *p* = 0.85). Conclusions: These data suggest that IPC can promote acute improvements in handgrip strength and functional capacity in active elderly women.

## 1. Introduction

Aging manifests as a complex, multidimensional phenomenon, with differences among individuals throughout life, and is highly conditioned by interactions among genetic, environmental, behavioral, and demographic characteristics [1]. Aging is a natural phenomenon that brings physiological changes, that can lead to physical and/or functional limitations, such as reduced muscle strength, agility, and cardiorespiratory fitness [2].

Sarcopenia, an age-related loss of muscle mass [3], has an estimated prevalence of 10% in adults over 60 years, increasing to 50% in adults over 80 years [4], and is associated with loss of muscle strength, functional capacity, and increased morbidity in elderly populations. In addition, muscle weakness is highly associated with mortality and with physical and functional disability [2]. Thus, it is increasingly important to identify evidence-based interventions that can mitigate the functional decline occurring progressively with advancing age. In recent years, strength training has been used in the elderly, and its benefits have gained increasing attention in the scientific community [5,6,7].

In light of this, more aggressive protocols of resistance training involving the addition of the so-called blood flow restricted (BFR) technique have also been trialed with success in the elderly [5,6,7]. Several studies [7,8,9] have demonstrated significant improvements in functional capacity in this deteriorated population after a period of strength training with BFR and low loads, compared to a control group. In this same context, another method of vascular occlusion, ischemic preconditioning (IPC), has been reported to be ergogenic for musculoskeletal recovery [10,11] and sports performance [12]. However, IPC has not yet been investigated in an elderly population. IPC is a method that alternates transient periods of complete vascular occlusion and reperfusion by applying a pneumatic tourniquet to the proximal region of the upper or lower limbs, at rest, before exercise, in order to precondition varied physiological functions [13,14].

IPC is an attractive method for athletes and exercisers due to its relationship with exercise performance [15]. Several studies have investigated the effects of IPC on swimming [16], cycling [17], and running performance [18]. In addition, studies have investigated the effect of IPC on resistance exercise [19,20,21], high-intensity interval training [22,23], and isometric exercise [13,24]. Considering that IPC promotes acute improvements in muscle strength in both males and females [14,19,21] and aerobic conditioning [18,22,25] in young people, we hypothesize that IPC could represent a relevant strategy to acutely increase muscle strength and, thereby, functional capacity in the elderly. Thus, the present study aimed to verify the acute effect of IPC on isometric handgrip strength and functional capacity in active elderly women.

## 2. Materials and Methods

### 2.1. Experimental Design

The present study (Figure 1) included four visits to the laboratory with a 72-h interval, performed at the same time of day (08:30–10:30 h) to reduce the circadian influence. During the first visit, the free informed consent (FIC) and Physical Activity Readiness Questionnaire (PAR-Q) were filled up and signed, and blood pressure, heart rate, and anthropometric measurements were evaluated. Then, the following tests were performed to assess baseline scores: the handgrip isometric strength test (HIST) and three functional tests (FT) which included a 30-s sit and stand. In the second, third, and fourth visits, the participants performed the same tests in a randomized crossover and single blind design with (a) IPC (see procedures below); (b) placebo compressions (SHAM); (c) alone (control protocol). The participants were instructed to abstain from exercise, avoid caffeine, chocolate, nutritional supplements, and alcohol intake in the 48 h preceding the collections, as well as to try to sleep at least six hours the night before the experimental sessions. During the session, the participants were instructed not to perform the Valsalva maneuver.

### 2.2. Sample and Ethical Procedures

Twenty elderly women who practiced aqua aerobics for at least 6 months (3 sessions/week; an average of 60 min/session) volunteered for the study (Table 1) [26]. The exclusion criteria were: (a) subjects who answered positively to any of the items of the PAR-Q [27]; (b) those who missed one of the sessions of the collection procedures in the laboratory; (c) those who presented some type of osteoarticular lesion in the upper or lower limbs and (d) smokers. The inclusion criteria were: a) the subjects had to be 60 years old or older, and (b) they had to have practiced aquarobics for at least 6 months. Five women who did not comply with the criteria for inclusion in the study were excluded, and 15 women completed the entire study. After the explanations about the risks and benefits of the research, the subjects signed the =FIC prepared according to the Helsinki declaration. This study was approved by the local Research Ethics Committee (n. 4.001.933/2020).

### 2.3. Procedures

#### 2.3.1. Anthropometric Measurements

Height and body mass were measured with a 0.5 cm precision stadiometer and a 0.1 Kg precision Filizola^®^ scale, respectively. All measurements were performed following the recommendations of ACSM [28]. These measurements were subsequently equated to obtain the body mass index (BMI) in kg.m^−2^. Muscle thickness and body fat percentage measurements were performed using BodyMetrix Pro (IntelaMetrix Inc., Livermore, CA, USA), following the recommendations suggested by Evangelista et al. [29]. Fat percentage was automatically calculated from Body View Professional software (IntelaMetrix Inc., Livermore, CA, USA) [30].

#### 2.3.2. Handgrip Isometric Strength Test and Functional Test Procedures

The HIST was evaluated using the Manual Dynamometry test with a Jamar dynamometer (capacity of 100 Kgf). After a warm-up, the volunteer was comfortably seated and positioned with the shoulder adducted, elbow flexed at 90°, forearm in the neutral position and, finally, the position of the wrist could vary from 0 to 30° of extension. The participant was then instructed to perform three attempts with the dominant hand, with a 1 min interval between them. The three attempt scores were then averaged [31].

FT were performed after the IPC and SHAM interventions and HIST. The 30s-chair stand (30STS) was performed as previously described by Jones et al. [32], the timed up and go (TUG) by Schaubert and Bohannon [33], and the 6-min walk test (6MWT) by the American Thoracic Society guidelines [34]. The 30STS test is related to the strength of the lower limbs, which evaluates the greatest number of correct repetitions of the person who sits down and stands up from a chair without the aid of arms in 30 s. The TUG test is related to balance and agility, in which the shortest time to get up from a chair, walk 2.44 m, go around a cone, and come back and sit on the chair again is measured. The 6MWT test is related to cardiorespiratory fitness, which evaluates the longest distance the volunteer can walk in 6 min on a 50-m track.

#### 2.3.3. Ischemic Preconditioning Protocol (IPC)

The IPC protocol consisted of 3 cycles of 5-min occlusion at 15 mmHg above systolic blood pressure (average: 132 ± 18.9 mmHg) separated by 5 min of reperfusion at 0 mmHg as per Jean St-Michael et al. [35] guidelines. A pneumatic tourniquet (57 cm × 9 cm, Riester^®^ komprimeter, Jungingen, Germany) was applied around the subaxillary region of the arm on both arms, and the compressions alternated from one arm to the other. The volunteers remained seated during the protocol to be checked for obstructed blood flow during the intervention with radial pulse manual checkup manually by digital palpation [14].

The SHAM protocol consisted of 3 cycles of 5-min occlusion at an absolute pressure of 20 mmHg, alternating with 5 min at 0 mmHg, as proposed in previous studies [14,20]. The volunteers remained seated at all times during the protocol.

During the control protocol, the volunteers remained seated for 30 min before performing the warm-up and after the tests.

### 2.4. Statistical Analysis

The normality of data sets was tested by the Shapiro-Wilk test. Results are presented as mean ± standard deviation. One-way ANOVAs with repeated measures and Tukey’s post-hoc test were used to test compare the means of the hand grip strength and functional capacity tests. The sample size was calculated using the software G* Power (ver. 3.1.9.7; Heinrich-Heine-Universität Düsseldorf, Düsseldorf, Germany). Based on a previous analysis, a sample of 15 individuals was calculated after using a power of 0.80, α = 0.05, a correlation coefficient of 0.5, the correction for non-sphericity of 1, and an effect size of 0.35. It was found that the sample size was sufficient to provide 81.8% of the statistical power. The procedures suggested by Beck [26] were adopted to calculate the sample. All analyses were performed in GraphPad software (Prism 6.0, San Diego, CA, USA), and an α value of 5% was considered.

## 3. Results

Results are displayed in Figure 2. The IPC induced a greater strength in the HIST (29.3 ± 6.9, *F*_(14, 28)_ = 46.42; *p* = 0.0001; eta^2^ = 0.946), compared with CON (27.3 ± 7.1; 7.1%; *p* = 0.01), but not with SHAM (27.7 ± 7.9; 5.5%; *p* = 0.16). The IPC induced a greater strength in the30STS (20.1 ± 4.1, *F*_(14, 28)_ = 17.81; *p* = 0.0001; eta^2^ = 0.866) compared with both SHAM (18.5 ± 3.5; 8.7%; *p* = 0.01) and CON (18.5 ± 3.9; 8.7%; *p* = 0.01). The IPC induced a greater strength in the TUG (5.70 ± 1.35, *F*_(14, 28)_ = 46.48; *p* = 0.0001; eta^2^ = 0.946) compared with SHAM (6.14 ± 1.37; −7.2%; *p* = 0.01), but not with CON (5.91 ± 1.45; −3.7%; *p* = 0.24). The IPC induced a greater strength in the 6MWT (611.5 ± 93.8, *F*_(14, 28)_ = 2.851; *p* = 0.0089; eta^2^ = 0.524) compared with CON (546.1 ± 80.5; 12%; *p* = 0.02), but not with SHAM (598.7 ± 67.6; 2.1%; *p* = 0.85).

## 4. Discussion

The present study aimed to verify the acute effect of IPC on hand grip isometric strength and functional capacity in active elderly women. The main findings were: (a) IPC significantly increased hand grip strength when compared to the CON protocol; (b) IPC significantly increased the number of repetitions in the 30STS test when compared to the SHAM and CON protocol; (c) IPC significantly increased performance in the TUG test when compared to the SHAM protocol; (d) IPC significantly increased performance in the 6MWT test when compared to the CON protocol. To the best of our knowledge, this study is the first to investigate the acute effect of IPC application on the elderly. Thus, the results of the current study provide unprecedented insight into the utility and efficacy of applying IPC for improvements in hand grip isometric strength and functional capacity in older people.

In our study, IPC increased hand grip isometric strength performance when compared to the CON protocol in active elderly women. According to Fragala et al. [2], hand grip strength is a relevant marker of muscle weakness (<16 kg in women) and is well established as a biomarker of age-related physical disability and early mortality [36,37]. In addition, recent recommendations to combat sarcopenia include the enhancement of muscle strength [38]. In previous studies, IPC had already demonstrated positive effects on isometric strength in healthy trained youth [19,24,39], which corroborates our current findings. For example, Libonati et al. [39] investigated the effects of IPC on the performance of rhythmic isometric wrist flexion in healthy young people and pointed out that IPC before exercise generated greater isometric strength compared to the CON group. Barbosa et al. [19] investigated the effects of IPC in delaying fatigue development in handgrip exercise. Results demonstrated a prolonged time to muscle failure, indicative of greater muscle endurance. In addition, Tanaka et al. [24] showed that IPC increased muscle endurance during isometric leg extension exercise. According to Tanaka et al. [24], the improvement in isometric and isokinetic muscle strength may be related to the increase in oxygen saturation and deoxygenation velocity of muscle hemoglobin and myoglobin concentration during exercise. On the other hand, Barbosa et al. [19] reported that the increase in hand grip isometric strength was not followed by the increase in muscular oxygen saturation during exercise.

Our findings also showed that IPC significantly increased repetition performance on the 30STS test when compared to the SHAM and CON protocols. The 30STS is a commonly used test to assess lower limb strength in elderly people [29,40]. In addition, the 30STS test is associated with levels of dynamic balance and cardiorespiratory endurance, and therefore represents the functional capacity of the elderly [40]. Some studies have reported that IPC promotes improvements in lower limb muscle endurance in recreationally trained youth [14,21]. Da Silva Novaes et al. [14] demonstrated that IPC with 4 cycles of 5-min occlusion at 220 mmHg increased the number of repetitions and total volume in a multi-articular resistance exercises session of the lower limbs and upper limbs when compared to the SHAM and CON protocols. Recently, da Silva Telles et al. [21] showed a positive effect of IPC on the number of repetitions for upper and lower limbs as a warm-up method for resistance exercise. Therefore, the current findings extend those previous data to show that the elderly could also benefit from IPC as much as younger populations.

The current study also reports for the first time the effects of IPC on TUG, and reports that transient blood occlusions can significantly increase performance on the TUG test when compared to the SHAM protocol. However, the IPC did not out perform the CON protocol. The TUG test is related to balance and agility in the elderly [41], and also provides a reliable measure of functional capacity in the elderly [42].

IPC also enhanced performance on the 6MWT test in these older women, which is related to aerobic capacity and functional ability in this target group [42]. It is also a risk indicator of all-cause mortality in the elderly [43]. Several studies have demonstrated the acute positive effect of IPC on aerobic capacity in many sports [25,44] corroborating our findings.

It is important to highlight that IPC was applied on arms in the current study and still generated improvements in the 30STS, TUG, and 6MWT tests, which mainly rely on lower limb muscle performance and aerobic capacity. Thus, the current data confirms that IPC produced systemic and remote ergogenic effects in elderly women. Previous studies have also demonstrated a remote effect of IPC application [14,19,20]. Barbosa et al. [19] demonstrated that IPC applied to the thighs increased strength during a hand grip isometric task. Marocolo et al. [20] evaluated the effects of IPC applied to the arms and thighs on upper limb resistance exercise performance in recreationally-trained men. The results of the study showed that IPC or SHAM, regardless of being applied to the arms or thighs, improved the performance in upper limb resistance exercises. Recently, Novaes et al. [14] also demonstrated that applying IPC to the arms before a resistance training session, alternating by segment and composed of six exercises for upper and lower limbs increased performance in repetitions and total training volume for lower limb exercises. Therefore, current and previous findings demonstrate that remotely applied IPC can acutely improve lower limb muscle performance and aerobic conditioning in active elderly women.

In our study, although IPC showed significant improvements compared to the CON protocol, it was not able to outperform SHAM in hand grip strength and 6MWT. de Souza et al. [45] point out that this improvement is not dependent on vascular occlusion and suggest that the SHAM maneuver may have generated a possible psychophysiological effect, influencing the central motor drive, which may have increased muscle performance. However, it should be noted that IPC is a method that needs further investigation.

Although we did not assess any physiological responses to exercise previous IPC studies have reported varied and complex physiological mechanisms that can explain the remote ergogenic effects observed here [12,46]. In fact, shear stress and local tissue hypoxia induced by the maneuver increase nitric oxide (NO) levels, a potent vasodilator [47], and activate vascular endothelial growth-factor (VEGF-α) gene expression [48]. Unsurprisingly, therefore, IPC has been associated with improvements in local vasodilation, blood flow [49] and, ultimately, O_2_ uptake [24], muscle Hb/Mb deoxygenation [13,22], and phosphocreatine and ATP stores [50]. These upregulated physiological responses could therefore explain part of the acute performance improvements in the HIST and FTs after IPC. Based on these performance test results, we could speculate that the IPC-induced upregulation in these vascular and metabolic functions also occurs in elderly women.

We can point out some limitations in our study: (a) we did not conduct direct measurements of vascular and metabolic variables related to performance (blood flow, blood lactate, muscle oxygen saturation, exercise VO_2_), which could clarify the mechanisms responsible for the increase in performance caused by IPC; (b) since we did not apply more IPC sessions, we cannot affirm that the ergogenic effects can last for more sessions and/or be translated into greater chronic adaptations on the variables analyzed. In this way, we recommend, in the future, more investigations about the effects of IPC on the elderly in different exercises, variables, physical abilities, clinical conditions, training levels, as well as the application of the IPC maneuver.

## 5. Conclusions

The application of three cycles of 5 min occlusion (with only 15 mmHg above systolic pressure) followed by reperfusion is effective in acutely increasing hand grip strength and performance in varied functional tests in active elderly women. It is a safe, low-cost, noninvasive, and easy-to-apply strategy to be used in elderly women who are in physical training programs. Thus, we recommend IPC to be applied before physical training or physical therapy in active elderly people by physical education professionals and/or physiotherapists in physical training and/or rehabilitation centers.

## Figures and Tables

**Figure 1 ijerph-19-06628-f001:**
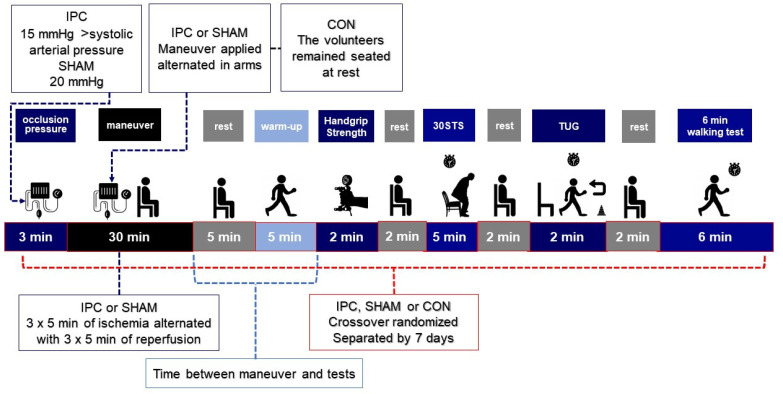
Experimental design; IPC: ischemic preconditioning; SHAM: placebo; CON: control protocol; TUG: timed up and go; 30STS: 30 s chair stand.

**Figure 2 ijerph-19-06628-f002:**
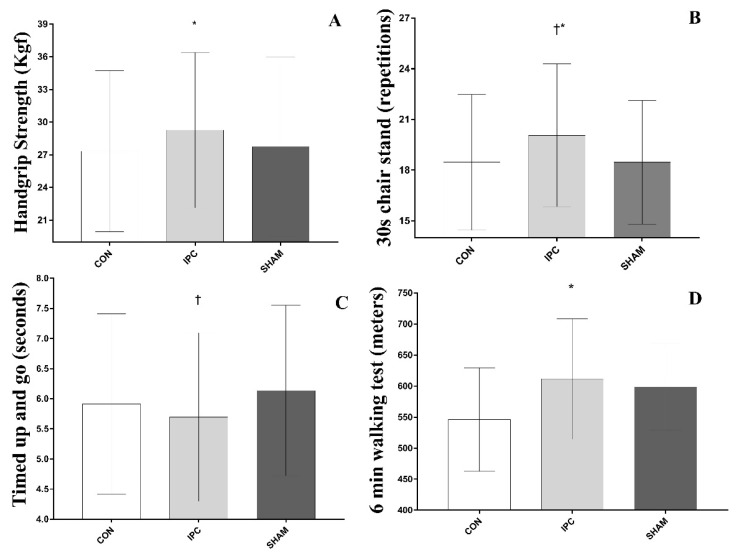
(**A**) handgrip strength; (**B**) 30 s chair stand; (**C**) timed up and go; (**D**) 6 min walking test * Significant differences with IPC vs.; † Significant differences with IPC vs. SHAM.

**Table 1 ijerph-19-06628-t001:** Characteristics of subjects (*n* = 15).

Age (years)	68.1 ± 7.6
Height (cm)	156.6 ± 9.1
Weight (kg)	70.5 ± 13.5
Body Fat%	32.2 ± 6.9
Muscle Thickness (mm)—(Biceps Bracchi)	23.2 ± 5.1
Muscle Thickness (mm)—(Rectus Femoris)	17.9 ± 3.9
Systolic Arterial Pressure (mmHg)	117 ± 18.9
Diastolic Arterial Pressure (mmHg)	75.1 ± 9.4
Heart Rate (bpm)	80.1 ± 12.3

## Data Availability

Available through the corresponding author by request.

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
