# Peer review of "Ischemic Preconditioning Improves Handgrip Strength and Functional Capacity in Active Elderly Women"

_ijerph, 2022, doi:10.3390/ijerph19116628_

Round 1

Reviewer 1 Report

The main problem detected is the excess of self-references. References 5, 14, 15, 16, 21, 23, 26 and 47 are from one of the authors. Although reference 16 is clearly justified (so much so that it diminishes the originality of the study, since they are almost identical studies, but for men), the rest seem to be self-citations.

We believe that the confounding variables should be clarified. Although in the methodology it is made explicit that the study subjects should not take caffeine, chocolate, etc., there is no explanation of the extent to which these variables could affect them, nor is there any explanation of how to get the study subjects to follow the indications of the authors. Likewise, the study is focused only on non-smoking women, so we consider that the title should reflect this characteristic.

It is imperative to show in the results the sample losses in the study. It is only indicated that 15 women have completed the study, but losses to follow-up are not indicated. 

The selection bias has not been clarified either. It cannot be ruled out that the study does not present a selection bias.

Graph 2 (results) could be improved. 

Line 249 indicates that the study focused only on functionality. It should be better justified (perhaps in the introduction), why these tests are chosen and not others. 

The study has many limitations, but it is good that the authors note this.

Author Response

Dear Reviewer,

We are grateful for your consideration of this manuscript, and we also very much appreciate your suggestions, which have been very helpful in improving the manuscript. We also thank the reviewers for their careful reading of our text. All the comments we received on this study of all reviewers have been attended into account in improving the quality of the article, and we present our reply to each of them separately.

The main problem detected is the excess of self-references. References 5, 14, 15, 16, 21, 23, 26 and 47 are from one of the authors. Although reference 16 is clearly justified (so much so that it diminishes the originality of the study, since they are almost identical studies, but for men), the rest seem to be self-citations.

A: Comment acknowledged and we agree with it. We have removed references 5, 26 and 47, but decided to keep ref 14 as it is relevant to the current study. Also, please note that ref 21 was not from our teams so we kept it in the manuscript.

We believe that the confounding variables should be clarified. Although in the methodology it is made explicit that the study subjects should not take caffeine, chocolate, etc., there is no explanation of the extent to which these variables could affect them, nor is there any explanation of how to get the study subjects to follow the indications of the authors. Likewise, the study is focused only on non-smoking women, so we consider that the title should reflect this characteristic.  These confounding variables can affect performance positively or negatively. This is common practice in sport science and exercise physiology research that subjects be asked to avoid these substances that can affect performance. Describing how such substances would affect performance is out of the scope of the study (and has been detailed elsewhere). The researchers only guided the volunteers about following the recommendations. But we couldn't know if everyone followed the guidelines as no dietary journal was used.

A: Yes, the Reviewer is correct that the present study has been performed with active elderly women (water aerobics practitioners) who were nonsmokers. This exclusion criteria was indeed been mentioned in the original manuscript, and we respectfully disagree with the Reviewer that exclusion criteria should be included in the title. We believe the title is precise and relevant to the content of the study.

It is imperative to show in the results the sample losses in the study. It is only indicated that 15 women have completed the study, but losses to follow-up are not indicated. 

A: We fully agree wit the Reviewer that this is a very important and relevant information. We have now mentioned in the manuscript that twenty women were recruited, but only fifteen completed the study.

The selection bias has not been clarified either. It cannot be ruled out that the study does not present a selection bias.

A: Our wording was not clear enough and this section has been clarified to mention that the order of the sessions was randomized. The section 2.1 now reads as follows: In the second, third and fourth visits, the participants performed the same tests in a randomized crossover and single blind design with a) IPC (see procedures below); b) placebo compressions (SHAM); c) alone (control

Graph 2 (results) could be improved. 

A: Comment acknowledge and this has been done.

Line 249 indicates that the study focused only on functionality. It should be better justified (perhaps in the introduction), why these tests are chosen and not others. 

A: We are going to remove this part, because the fact that we did not focus on measuring the physiological variables was due to lack of resources, equipment and logistics in our laboratory, so we decided to verify with tests that were within our reach and could reflect in a good practical application for our study.

The study has many limitations, but it is good that the authors note this.

Reviewer 2 Report

In a single-blind, placebo-controlled study the authors aimed to verify the acute effect of IPC on handgrip strength and functional activity in active elderly women. The study is of great clinical significance. It is the first to investigate the issue.

The experimental design is clear, the tests used (HIST, 30STS, TUG and 6MWT) are simple and easy to evaluate and analyse. The results provide promising therapeutic option to the clinical practice, as all parameters examined significantly improved compared to either Con or SHAM group or both.

Figures and tables are informative and easy to follow. Statistical analysis is correct.

Some remarks:

No information was provided about any diseases of the participants.

The manuscript did not mention whether participants felt discomfort or pain during IPC.

There is no information reported on the effects of IPC on other body systems (cardiopulmonary, vascular etc). However, this limitation and some others were described also by the authors themselves.

To set up a “gold standard” protocol further research is needed to investigate the ideal pressure, duration of occlusion and number or cycles.

Finally a small note: the explanations of the abbreviations FIC and PAR-Q mentioned in section 2.1 can only be found in section 2.2

Author Response

Dear Reviewer

We are grateful for your consideration of this manuscript, and we also very much appreciate your suggestions, which have been very helpful in improving the manuscript. We also thank the reviewers for their careful reading of our text. All the comments we received on this study of all reviewers have been attended into account in improving the quality of the article, and we present our reply to each of them separately.

In a single-blind, placebo-controlled study the authors aimed to verify the acute effect of IPC on handgrip strength and functional activity in active elderly women. The study is of great clinical significance. It is the first to investigate the issue.The experimental design is clear, the tests used (HIST, 30STS, TUG and 6MWT) are simple and easy to evaluate and analyse. The results provide promising therapeutic option to the clinical practice, as all parameters examined significantly improved compared to either Con or SHAM group or both.

Figures and tables are informative and easy to follow. Statistical analysis is correct.

Some remarks:

No information was provided about any diseases of the participants

The exclusion criteria mentioned that participants who answered positively to any of the items of the Physical Activity Readiness Questionnaire / PAR-Q were excluded. Therefore, we did not report possible diseases, as the PAR-Q excludes these people from the study.

The manuscript did not mention whether participants felt discomfort or pain during IPC.

A: Thank you for this comment. Unfortunately, we did not measure pain sensation in our study, but we asked about discomfort. 

There is no information reported on the effects of IPC on other body systems (cardiopulmonary, vascular etc). However, this limitation and some others were described also by the authors themselves.

A: Yes, this study was focused on muscle strength, and we have included this limitation.

 To set up a “gold standard” protocol further research is needed to investigate the ideal pressure, duration of occlusion and number or cycles.

Finally a small note: the explanations of the abbreviations FIC and PAR-Q mentioned in section 2.1 can only be found in section 2.2

A: Thank you. These have now been defined in section 2.1.